# Pesticide Residues in Bananas from the Canary Islands

**DOI:** 10.3390/foods12030437

**Published:** 2023-01-17

**Authors:** Juan M. Méndez, Ángel J. Gutiérrez-Fernández, Arturo Hardisson, Daniel Niebla-Canelo, Samuel Alejandro-Vega, Carmen Rubio-Armendáriz, Soraya Paz-Montelongo

**Affiliations:** 1Area of Toxicology, University of La Laguna, 38071 La Laguna, Canary Islands, Spain; 2Grupo Interuniversitario de Toxicología Alimentaria y Ambiental, Universidad de La Laguna, 38071 La Laguna, Canary Islands, Spain

**Keywords:** pesticides, banana, Canary Islands, toxic risks

## Abstract

There are a large number of pests which are detrimental to plant production, specifically to banana cultivation, and the use of pesticides is the main method of control of these pests. Therefore, the number of active substances in pesticides has been steadily increasing since before the Second World War. There is growing consumer concern about the health effects of pesticide residues and there is certainly evidence of a link between pesticide exposure and the prevalence of chronic diseases. Therefore, it is of particular interest to study the presence of pesticide residues in bananas and their toxicological, agricultural and legal implications. In this study, the content of pesticide residues in bananas produced in the Canary Islands during a ten-year period from 2008 to 2017 was determined. A total of 733 samples of bananas were analysed during the study period, in which 191 different active substances were investigated, involving 103,641 assessments. The samples analysed were selected in such a way that they are representative of the banana sector in the Canary Islands as a whole, taking into account geographical and climatic factors, cultivation methods and the processing of bananas in packaging, which are the differentiating factors in the use of pesticides. The reference parameter for the residue investigation is the MRL (maximum residue limit).

## 1. Introduction

The Food and Agriculture Organisation of the United Nations (FAO) maintains that plant pests and diseases affect crops, causing significant losses in agricultural production. The spread of plant pests and diseases has markedly increased in recent years and has become a transboundary phenomenon. Pests can easily spread from one country to another and reach epidemic proportions, causing substantial losses in agriculture, endangering farmers’ livelihoods and the food security of millions of people [1].

The European Union (E.C. Regulation 396/2005) [2] states that the production and consumption of plant and animal products are of great importance in the community. The use of plant protection products is one of the most important ways to protect plants and plant products against harmful organisms and to improve agricultural production (E.C. Regulation 1107/2009) [3]. On the other hand, people can be exposed to pesticides both directly, e.g., through occupational activities such as pesticide application in agriculture, and indirectly through drinking water, air, dust and food. Although the pesticides that affect the population are mainly from agricultural use and are responsible for long-term exposure generally at low concentrations, they are also found in the environment [4].

Pesticides are biologically active substances which are toxic to living organisms and which may have consequences for human health [5]. This toxicity can be acute or chronic. The toxic agent propagates through toxicodynamic processes to the manifestation of toxicity at the organismal level. Exposure in turn is a function of dose and time [6,7]. The association between pesticide exposure and the development of chronic diseases in humans is difficult to demonstrate [4].

The only research on the pesticide residue levels in bananas from the Canary Islands found in a search of the literature is the study by Hernández-Borges et al. [8], conducted between 2007 and 2008. This article showed that the most frequent pesticides used on bananas are chlorpyrifos (88% of the samples).

Another study by Gomes et al. [9] reviewed fifteen articles on the presence of pesticides in bananas. According to Brazilian legislation, 34.6% of the samples presented residues > MRL and 65.4% of the samples presented an unauthorised active ingredient. According to EU legislation, 32.4% of the samples showed residues > MRL and according to Codex Alimentarius standards, 42.6% of the samples did not comply with their acceptance criteria.

The European Union has been carrying out extensive monitoring of pesticide residues in food under the coordination of the European Food Safety Authority (EFSA), the results of which have been set out in two annual reports “European Union Report on Pesticide Residues in Food. European Food Safety Authority” (2008–2017) [10,11,12,13,14,15,16,17,18,19]. These programmes are underpinned by successive EU Regulations which regulate them [20,21,22,23,24].

The controls on pesticide residues carried out by the EU are set out in two types of programmes, the “Coordinated EU Programme” and the “National Control Programmes”. The aims of these programmes are to analyse randomly selected samples to collect data on the presence of pesticides in representative fruits, vegetables and cereals for the European market, which are suitable for assessing the actual dietary exposure of the European population.

Eighteen active substances were detected, and of these, eleven had concentrations > LOQ but < MRL. Thiabendazole, imazalil and chlorphyrifos were repeatedly found every year in a significant % of the samples. Azoxystrobin was also detected in a considerable percentage of the samples, but only in the years 2009, 2012 and 2015. Bifenthrin, buprofezin, fenpropimorph, dithiocarbamates and mycobutanil were detected in a lower percentage of samples (between 3 and 6%) but only in some years. Twelve active substances were detected with a concentration > MRL, but always in a low % of samples [10,11,12,13,14,15,16,17,18,19].

## 2. Materials and Methods

The research method consisted of taking samples of bananas ready for marketing and analysing them for the presence of pesticide residues, determining whether or not they exceeded the MRL set for each active ingredient.

### 2.1. Sampling

The samples were taken from packets of bananas ready to be marketed. Therefore, the samples were taken from the bananas at the point they reach the consumer. On the other hand, it should be noted that Commission Regulation (EU) No 1333/2011 [25], lays down marketing standards for bananas and defines the qualities that unripe green bananas must have once they have been packed and packaged. Therefore, samples for pesticide residue analysis were taken at the same stage, i.e., green bananas, so that the sampling coincided with the time when the bananas had to meet the marketing requirements.

The samples were collected taking Into account the factors that may influence the presence of pesticide residues, in order to try to establish a relationship between the presence of pesticides and these factors, which are as follows: origin, the producer organisation, the harvesting month, the climatic factors and the category.

Table 1 shows the number of samples taken from each island, distributed over the ten years of the study.

The month of banana production and harvesting may be related to the number and quantity of pesticide treatment applied to the crop, and the rainfall, relative humidity, sunshine and ambient temperature, which are characteristic of the different seasons, may determine the presence of the different pests that affect this crop. In the Canary Islands, two main climatic zones can be distinguished which have different characteristics, namely the northern and southern slopes of the islands, which are more homogeneous than those found within each island.

Table 2 shows the distribution of the samples taken per month over the years during which the research was conducted. Sampling remained acceptably uniform throughout the year, except for the months of January and December, when sampling only represented 5% and 3% of the samples taken, corresponding to thirty-five and twenty samples, respectively.

Bananas can be classified into Extra, First and Second Class, as described in the legislation (Regulation (EU) No 1333/2011) [25]. In order to classify a consignment of bananas into one category or another, a series of requirements must be met, which are related to the number and intensity of defects that can be admitted in each one:−Extra Class: Bananas in this class must be of superior quality. They must be characteristic of the variety or commercial type. Fingers must be free of defects, with the exception of very slight superficial defects, not exceeding 1 cm^2^ of the total surface area of the finger, provided these do not affect the general appearance of the individual hand or bunch, the quality, storage quality and presentation in the package;−Class 1: Bananas in this class must be of good quality. They must be characteristic of the variety or commercial type. The following slight defects, however, may be allowed provided these do not affect the general appearance of the hand or bunch, the quality, the keeping quality and presentation in the package: slight defects in shape; slight skin defects due to rubbing; and other superficial defects, not exceeding 2 cm^2^ of the total surface area of the finger. In no case may the slight defects affect the flesh of the fruit;−Second Class: This class includes bananas which do not qualify for inclusion in the higher classes, but satisfy the minimum requirements specified for the class. The following defects may be allowed, provided the bananas retain their essential characteristics as regards the quality, the keeping quality and presentation: defects in shape and skin defects due to scratching, rubbing or other causes, not exceeding a total of 4 cm^2^ of the surface area of the finger. Under no circumstances may such defects affect the flesh of the fruit.

In addition to the existing categories (Extra, First and Second Class), non-categorised samples were also collected because they were taken before the washing, conditioning and final treatment before packaging. This is a characteristic group in itself, because certain final treatments have not yet been applied at this stage.

Table 3 shows the list of samples taken, distributed by category, throughout the study. It can be seen that 40% of the samples correspond to the First Class Category (290 samples); 37% to the Extra Class (272 samples); 18% to non-categorised samples (131 samples); and 5% to the Second Class category (40 samples). Ideally, the same number of samples would have been taken from each category, but this was not possible because the sampling was during normal packing activity, at the expense of the categories being packed at the time. Even so, the forty samples in the Second Class category are sufficient to obtain a meaningful result.

### 2.2. Sampling Procedure

The samples were taken from packs of bananas previously selected as being representative of the sector according to the criteria explained above. Two samples were taken from each of two different commercial categories (Extra, First and Second Class and non-categorised). In order to do this, the bananas present in the packages were separated by choosing two of them at random.

The sampling procedure was carried out in accordance with the provisions of the Royal Decree 290/2003 in Spain [26], establishing the sampling methods for the control of pesticide residues in products of plant and animal origin, and of the Royal Decree 280/1994 in Spain [27], establishing the maximum residue limits of pesticides and their control in certain products of plant origin, and in general in accordance with the provisions of Regulation 625/2017 [28], on controls and other official activities carried out to ensure the implementation of food and feed law.

In the case of bananas, ten units must be taken, which must weigh at least 1 kg (Royal Decree 290/2003), so each sample taken weighed 1 kg and contained ten bananas.

The part of the plant products that have to comply with the MRL is indicated in ANNEX I of Royal Decree 280/1994. In the case of bananas, this is a whole green banana, with peel and without stalk. Therefore, the analysis was carried out on ten whole pieces with peel.

In turn, each of the samples taken consisted of three homogeneous samples for initial, contradictory and conclusive analysis, in accordance with the provisions of Royal Decree 1945/1983 [29], regulating infringements and penalties in the area of consumer protection and agri-food production, thus fulfilling the legal requirements established for sampling.

In addition, all sampled lots were checked for conformity with the marketing standards laid down in Regulation (EU) No 1333/2011 [25] and in Annex V (2) of Regulation (EU) No 543/2011 [30], laying down detailed rules of application of the fruit and vegetables and processed fruit and vegetables sectors. This ensures that the samples taken comply with the requirements for marketing.

Finally, the samples were bagged, labelled and sealed in such a way as to ensure traceability.

### 2.3. Number of Analysed Samples and Pesticides

A total of 733 samples were taken in the study period (2008–2017). The number of analysed samples by year were 52 samples (2008), 79 samples (2009), 43 samples (2010), 75 samples (2011), 62 samples (2012), 89 samples (2013), 19 samples (2014), 80 samples (2015), 86 samples (2016) and 88 samples (2017).

A total of 191 active substances were analysed over the study period. The number of active substances analysed each year is shown in Table 4, which distinguishes the number of active substances analysed by LC-MS/MS (liquid chromatography–mass spectrometry), GC-MS/MS (gas chromatography-mass spectrometry) and dithiocarbamates.

### 2.4. List of Analysed Pesticides

The number of determinations performed by LC MS/MS was 70,249; by GC MS/MS it was 33,391 and 733 dithiocarbamates, i.e., a total of 103,641 determinations during the whole study.

The active substances investigated are a sum of those authorised for bananas in the Register of Plant Protection Products of the Spanish Ministry of Agriculture, Fisheries, Food and the Environment; those authorised for other plant products such as avocados, mangoes and papayas, which constitute the majority of plantations in the Canary Islands and are therefore the most commonly used active substances; the list of active substances most frequently marketed in the Canary Islands, according to information obtained from the Directorate-General for Agriculture of the Ministry of Agriculture, Livestock and Fisheries of the Government of the Canary Islands; the list of active substances which have recently been prohibited and of which it is suspected that there are still stocks held by both farmers and supplying companies; and finally, the multi-residue method offered by the laboratories.

In principle, a basic multi-residue package was analysed, and each year, or on an individual basis, active substances that were of particular interest at a given time were analysed.

The methods used are valid for the analysis of pesticide residues in vegetables as they are methods recommended by reference laboratories and comply with the legal requirements established in the European Union.

Whole bananas without stalks were analysed in accordance with the provisions of ANNEX I of the Royal Decree 280/1994 of 18 February 1994, which establishes the maximum residue limits for pesticide residues and their control in certain products of plant origin, defining the plants and parts of plants to which it applies.

The sample weight was 1 kg and contained at least 10 units. The samples were crushed and homogenised until a homogeneous slurry was obtained and an aliquot of 100 g was taken. The extraction, purification and storage procedure that corresponds to each procedure was applied. Then, pesticide residues were determined.

### 2.5. Determination of Pesticide Residues by GC-MS/MS

Extraction was the first step of the test procedure. Pesticide residues were extracted from the sample in a two-step process: a first extraction step with acetonitrile using the QuEChERS (quick, easy, cheap, effective, rugged and safe) method and a second clean-up step using primary/secondary amines (PSA) for the removal of organic acids and polar pigments and other products.

An amount of 15.0 ± 0.05 g was taken from each previously crushed and homogenised sample. This was then subjected to an extraction process as follows: (1) 15 mL of acetonitrile with 1% glacial acetic acid was added, (2) this was subjected to 4 min agitation (Agytax, Madrid, Spain), (3) the salts of the extraction kit for QuEChERS (MgSO_4_, NaCl and anhydrous sodium acetate) were added, (4) this was subjected to manual agitation and then agitation by ultrasound (3 min), (5) it was centrifuged (4000 rpm, 5 min) and the supernatant was taken and transferred to SPE Dispersive Kit tubes (PSA and MgSO_4_) for purification, (6) it was shaken again for 1 min and centrifuged (4000 rpm, 2 min), (7) 1. 8 mL cyclohexane (Sigma Aldrich, Darmstadt, Germany) was added, followed by mechanical shaking (1 min) and ultrasonic shaking (1 min) and (8) an aliquot of the extract was taken, filtered and collected directly into the chromatography vial. The following equipment was used: a Varian GC3200 and Varian 450-GC gas chromatograph (Palo Alto, CA, USA), a Varian MS 240 and Varian 4000 MS mass spectrometer (Palo Alto, CA, USA), a Varian model 1079 PTV injector (Palo Alto, CA, USA) (programmable temperature, ramp) and a Varian 8400 autosampler (Palo Alto, CA, USA).

### 2.6. Determination of Pesticide Residues by LC-MS/MS

The following equipment was used: a Varian 320 MS TQ chromatograph, a Varian 212-LC pump and a Varian 410 Autosampler (Varian (Palo Alto, CA, USA)).

An ACE-C18-AR (10 cm × 2.1 mm; 3 µm) reverse phase column was used under the following chromatographic conditions: thermostated at 40 °C ± 3 °C; mobile phase A: 5 mM ammonium formate in water with 0.2% formic acid; mobile phase B: methanol; a flow rate of 0.2 mL/min; and an injection volume of 10 µL (in µpickup mode) (Advanced Chromatography Technologies Ltd. Reading, UK).

The quantification of the residue of each pesticide, as in the GC-MS/MS test procedure, was performed by interpolation of the peak area obtained for the quantification ion/ions of the pesticide, by the corresponding calibration curve.

### 2.7. Chromatography Conditions

Column: capillary, 30 m, 0.25 mmLD, 0.25 m phase (5% phenyl methyl polysiloxane or equivalent). Stationary phase: a capillary column with a stationary phase of the type 5% phenyl-methyl polysiloxane or an equivalent phase was used. Injection: split/splitless injection of large volumes, with a programmable temperature ramp (PTV) in the injector. Volume of sample injected: 10 microlitres. The columns used were from the following commercial brands: Agilent Vf-5 ms, Agilent HP-5 and Agilent DB-5 (Agilent (Santa Clara. CA, USA)).

### 2.8. Criteria for Identification and Confirmation

The identification and confirmation criteria were retention time, identification of the spectrum of a precursor ion and fragment ions. Among these ions there must be at least one confirmatory ion. Quantification of the residue of each identified and confirmed pesticide in the sample was performed by interpolation of the peak area obtained for the quantitation ion/ions of the pesticide in the corresponding calibration curve. The concentration of each identified and confirmed pesticide was calculated according to the following expression:Concentration = (Ap − b)/m(1)
where concentration is the concentration in ppb of the pesticide; Ap is the peak area for the quantification ion; b is the independent term of the calibration curve; and m is the slope of the straight line (Ap = concentration m + b).

## 3. Results and Discussion

Table 5 shows the summary of the presence of pesticides in samples from the study period. These data show a high number of samples with residues, around 95% on average, of which fifteen had residue concentrations above the MRL, accounting for 2.05% of the total. ADL is the analytical detection limit; N is the number of samples; and R+ is the number of samples with residues > ADL and residues < MRL.

The number of samples with pesticide residue concentrations above the MRL varies throughout the study. The years 2008 and 2009 show notably high values of 13.5% and 7.6%, respectively. These values decrease radically in the following years, an observation that coincides with the intensification of the monitoring of pesticide residues by the authorities in charge of agri-food quality control since 2007, which suggests a better rationalisation of pesticide use by operators.

Only two samples were detected from 2010 to 2017 with a residue concentration above the MRL, i.e., only 0.35% of the total of 567 samples analysed in that period, while in the last five years of the study (years 2013–2017) no residues were detected with concentrations above the MRL.

The controls carried out by the EU within the European Coordinated Programme are directly comparable, because they are standardised controls throughout the EU, conducted according to standardised test procedures and estimation of measurement uncertainty, which means that the results obtained by the different EU member states are directly comparable.

The number of active substances analysed in the Coordinated Community Programme has progressively increased from 78 samples taken in 2008 to 156 and 157 samples taken in 2015 and 2016, respectively, which is fully in line with the number of active substances analysed in the present study, ranging from 126 in 2008 to 162 in 2017. However, when comparing the data from the study here with those obtained in the Coordinated Community Programme, it can be seen that both are similar. In the Community Programme, the presence of pesticide residues with a concentration > MRL shows a decreasing variation in the period studied (2006–2015): 0.3% in 2005, 1.4% in 2006, 0.4% in 2009 and 0.7% in 2012. In the present study, except for the years 2008 and 2009 when the presence of residues is clearly higher than in the EU controls, the values are equal from 2010 onwards (0.35% Canary Islands/0.3% EU). During the years 2013 to 2017, no residues exceeding the MRL were detected in the Canary Islands in the total of the 422 samples analysed in that period, with the conclusion that bananas produced in the Canary Islands from 2013 onwards show a total control of the presence of pesticide residues above the MRL.

While the number of samples with pesticide residues above the MRL has decreased dramatically since 2010, the number of samples with residues below the MRL remained virtually unchanged (Table 5). It should be noted that food legislation allows for the presence of pesticide residues at concentrations below the MRL [31]. The data seem to suggest that, following the intensification of controls by the responsible authorities, operators have adjusted their processing to comply with the law. This improvement is based on the adjustment of operators to producing bananas with a lower residue content to comply with the regulations.

When comparing these data with those from the European Coordinated Programme, one can see that the presence of pesticide residues shows a decreasing concentration in the period of time studied (2006–2015), so the percentages of samples with concentrations > LOQ and <MRL were as follows: 55% in 2006; 56.8% in 2009; 77.8% in 2012; and 73.1% in 2015, significantly lower than the results obtained in the Canary Islands, where 94% of the samples show pesticide residues > LOQ and <MRL.

In order to evaluate the presence of pesticide residues in bananas from the Canary Islands in a more explanatory way, it is proposed here to relate this to the number of determinations carried out rather than to the number of samples analysed. Thus, of the total number of determinations performed in the present study (103,641), 1.54% of these (1595) had pesticide residues with a concentration higher than the analytical detection limit (ADL) but lower than the MRL, while the number of instances in which the MRL was exceeded was fifteen, 0.014% of the total, which indicates a very low incidence of residues in this product.

### 3.1. Number of Active Substances per Sample

A highly relevant piece of data is the number of active substances found in each of the samples. Figure 1 shows this information (for active substances with concentrations < MRL). It can be seen that most of the samples show residues of multiple active substances. Of the 695 samples in which residues were detected, on average, 24% had residues of one active substance, 40% of two active substances, 23% of three active substances and 10% of four active substances.

On the other hand, the sum of all the residues of active substances detected in the 733 samples analysed during the study period was 1606, which means a mean average of 2.2 residues per sample. The sum of the residues exceeding the MRL, out of the 733 in the same period, was fifteen, which means a mean average of 0.02 residues > MRL per sample. Only one sample was detected with two residues > MRL. This information is shown in Table 6.

### 3.2. Detected Active Substances

It is of interest to study which active substances were found to have residues. In total, 191 different active substances were analysed, but only thirty-six of them were found to have pesticide residues. Table 7 lists the active substances for which residues were found. Only four of these thirty-six substances showed residues in a concentration above the MRL. These limits are established in European legislation by means of regulations, and are therefore obligatory throughout the European Union. These limits are frequently revised by varying their reference concentration according to updated scientific evidence.

The active substances found can be classified according to the frequency with which they are detected. Substances such as chlorpyrifos and imazalil which were quantified in 71.2% and 68.5% of the samples, respectively. Substances such as buprofecin, indoxacarb, acetamiprid and hexythiazox are the next largest group, which were detected in between 11% and 14% of the samples. Substances such as bifenthrin, spirodiclofen, etoxazole, fenazaquin, lambda-cyhalothrin, clofentezine and imidacloprid were detected in between 3% and 5% of the samples. The rest of the active substances detected were found in a lower percentage of samples.

These data coincide with the data on chlorpyrifos reported in the article by Herández-Borges et al. [8], in which this active substance was found in 88% of the samples. The rest of the active substances detected were also detected in the present investigation, although in different proportions, which could be explained simply by the difference in the size, both in the number of samples and geographically, of the two studies.

A comparison of the results of the present research with the EU Coordinated Pesticide Monitoring Programme shows that fewer active substances were detected in bananas marketed in Europe, i.e., twelve compared to the thirty-six found in the present study. Chlorphyrifos, which was the main active ingredient detected in the Canary Islands, was also found in bananas marketed in Europe, but in a lower percentage of between 45 and 50%, compared to 68% detected in the Canary Islands.

Imazalil, which was the second most frequent active ingredient found in the Canary Islands, was also found in bananas marketed in Europe, but in a lower percentage of between 10 and 20%, compared to 71% detected in the Canary Islands. Thiabendazole, which was the third most frequent active ingredient in bananas marketed in Europe, was found in between 40 and 55% of the samples, compared to 1.2% found in the Canary Islands.

Substances such as bifenthrin and buprofezin were found in both bananas marketed in Europe and in the Canary Islands in similar percentages. For the rest of the active substances, there were slight differences between the two control programmes. However, there is a coinciding pattern between both studies, which is that the pesticide treatment of bananas and plantains is based on the combined use of insecticides, acaricides and fungicides.

### 3.3. Association of the Active Substances Found

A total of 103,641 determinations were performed in the present study, of which 1595 corresponded to determinations with a concentration of residues below the MRL (1.54% of these), while fifteen determinations detected a concentration above the MRL (0.014%). Of these 1596 determinations, 94% (1510) corresponded to only twelve active substances, which are listed in Table 8.

It should be noted that chlorpyrifos was detected in 71.2% of the determinations and imazalil in 69.5% of these, with these being the two most frequent active substances of all those detected. There is another group of four active substances (indoxacarb, acetamiprid, buprofecin and hexythiazox), which were detected in between 13.5 and 11% of the determinations. Finally, another group of six active substances (clofentezine, fenazaquin, imidacloprid, bifenthrin, lambda-cyhalothrin and spirodiclofen) were detected in 3 to 5% of the determinations.

The associations between these active substances were studied, with the result that chlorpyrifos and imazalil were associated in 46.9% of the samples analysed. Indoxacarb was frequently associated with chlorpyrifos and imazalil (between 8 and 10% of the samples) and with acetamiprid, buprofecin, hexythiazox, clofentezine and spirodiclofen (between 1 and 2.7% of the samples). Acetamiprid was mainly associated with chlorpyrifos (10% of samples), with imazalil (7% of samples) and with acetmiprid (3.5% of samples).

Buprofecin was mainly associated with chlorpyrifos (10.4% of samples), with imazalil (7.2% of samples) and with indoxacarb (3.5% of samples). Hexythiazox was mainly associated with chlorpyrifos (9.0% of samples) and imazalil (8.6% of samples).

There were also significant associations of three or more active substances. Chlorpyrifos, imazalil and indoxacarb were found to be associated in fifty-one of the samples analysed, i.e., 7% of the samples. Chlorpyrifos, imazalil and buprofecin were found to be associated in forty-three of the samples analysed, i.e., 5.9% of the samples analysed. Chlorpyrifos, imazalil and acetamiprid were associated in twenty-nine of the samples analysed, i.e., about 4% of the samples analysed. Chlorpyrifos, imazalil and hexythiazox were found in twenty-nine of the samples analysed, i.e., 2.3% of the samples. In general, the active substances were found have an insecticidal, acaricidal, fungicidal or larvicidal effect, so that it can be concluded that they are in accordance with the needs of the crop, i.e., no inappropriate active substances were detected.

The association of chlorpyrifos and Imazalil was the most frequent combination, detected in 46.9% of the samples, and is due to the pesticide control of insects and fungal diseases that affect this crop in the post-harvest phase. Therefore, this association is considered appropriate for the intended purpose. The association of chlorpyrifos and indoxacarb, which was detected in 10.6% of the samples, is also logical. Although both are insecticides, indoxacarb has a marked action against caterpillar and lepidopteran larvae, so it is logical that a combined treatment of both insecticides was sometimes chosen.

On the other hand, chlorpyrifos, which is a non-systemic insecticide, was detected in 29% of the samples associated with other insecticides and insecticide-acaricides (indoxacarb, acetamiprid, buprofecinb, imidacloprid and bifenthrin), which seems to be a notably high frequency of use of several insecticides in the same sample. In principle, it would not seem logical to use other insecticides together with chlorpyrifos, as the latter has a proven efficacy against the main insects affecting this crop. It may make sense to use it in combination with other systemic insecticides, such as acetamiprid or imidacloprid, as this achieves a broader effect on insect populations that are difficult to reach by contact. It may also seem logical to use chlorpyrifos with other active substances that have an acaricidal effect as well as an insecticidal effect, such as buprofecin, hexythiazox and bifenthrin. However, chlorpyrifos was also detected in association with other active substances that only have a non-systemic insecticidal effect, such as bifenthrin or lambda-cyhalothrin. In these cases, it seems that several insecticides with the same effect were being used, duplicating the use of these products without achieving a greater insecticidal effect; therefore, optimisation of and reduction in the use of pesticides was not achieved.

Imazalil, which is a systemic fungicide, was mainly associated with chlorpyrifos—an association already discussed in previous paragraphs—but it was also significantly associated with indoxacarb, acetamiprid and buprofecin in 26% of the samples. This combination seems logical, as it is due to the pesticide control of insects and fungal diseases that affect the post-harvest stage of this crop. Therefore, this association is considered appropriate for the intended purpose.

In the case of indoxacarb, which is a non-systemic insecticide and larvicide, as has already been discussed and explained, its association with chlorpyrifos and imazalil seems logical; however, it was also associated with other insecticides such as acetamiprid, buprofecin and hexythiazox. In these cases, it seems that two insecticides with the same effect were being used, duplicating the use of these products without achieving a greater insecticidal effect, again not optimising or reducing the use of pesticides.

Acetamiprid, present in 11.6% of the samples, is a systemic insecticide and was mainly associated with chlorpyrifos, imazalil and indoxacarb. The association with imazalil is clearly appropriate to combat the type of pests affecting this crop. In the case of the association with chlorpyrifos and Indoxacarb, although all three are insecticides, this can also be considered appropriate because a broader effect on insect populations that are difficult to reach with a locally acting insecticide alone may be pursued.

Buprofecin, present in 11.2% of the samples, has an acaricidal and non-systemic insecticidal effect. It was mainly associated with chlorpyrifos (10.4% of the samples), imazalil (7.2% of the samples) and indoxacarb (3.5% of the samples). While the association with imazalil seems appropriate to combat the type of pests affecting this crop, the association of buprofecin with other active substances which are also insecticides and combat the same pests does not indicate a consideration of the active substance of choice, as the pesticidal effect is being duplicated.

Hexythiazox, present in 11.2% of the samples, has a non-systemic insecticidal and acaricidal effect. It was mainly associated with chlorpyrifos (9.0% of the samples) and imazalil (8,6% of the samples). As already mentioned with other similar products, the fact that it was associated with a fungicide such as imazalil is logical because of the nature of the pests affecting the crop, but it is not considered appropriate that it is associated with chlorpyrifos, as both are active against the same type of pests.

## 4. Conclusions

This study is a comprehensive study in which the number of samples and the number of active substances investigated are appropriate to the size of the Canary Islands and comparable with the main pesticide monitoring programme carried out in Europe in the same period.

In general, the active substances found have an insecticidal, acaricidal, fungicidal or larvicidal effects, so it can be concluded that they are in accordance with the needs of the crop, i.e., no inappropriate active substances were detected. In some cases, it was found that several insecticides with the same effect were used, duplicating the use of these products without achieving a greater insecticidal effect; therefore, the optimisation of and reduction in the use of pesticides was not achieved.

In summary, it can be concluded that pesticide residues in bananas produced and marketed in the Canary Islands are well-controlled, with very few residues above the MRL. This finding is evidence showing that good agricultural practices are being used in this crop. However, it is necessary to maintain the monitoring of the use of pesticides.

## Figures and Tables

**Figure 1 foods-12-00437-f001:**
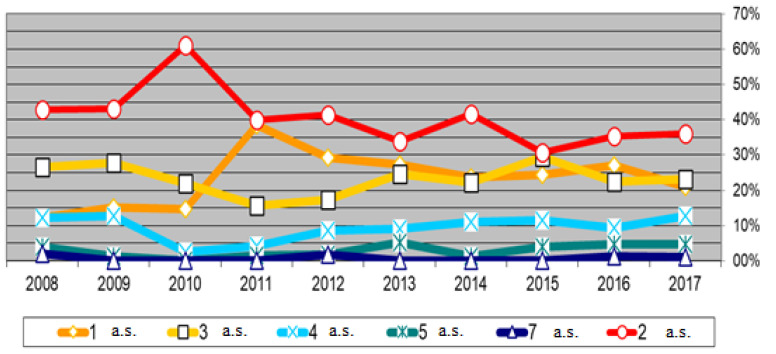
Number of active substances (a.s.) per sample.

**Table 1 foods-12-00437-t001:** Distribution of banana samples taken by island and year.

Islands	2008–2017		2008		2009		2010		2011		2012		2013		2014		2015		2016		2017	
	No	%	No	%	No	%	No	%	No	%	No	%	No	%	No	%	No	%	No	%	No	%
TENERIFE	370	50.5	24	46.2	37	46.8	35	81.4	33	44.0	36	58.1	34	38.2	47	59.5	39	48.8	45	52.3	40	45.5
LA PALMA	208	28.4	16	30.8	21	26.6	0	0.0	26	34.7	16	25.8	32	36.0	20	25.3	22	27.5	27	31.4	28	31.8
GRAN CANARIA	109	14.9	6	11.5	14	17.7	8	18.6	12	16.0	8	12.9	14	15.7	8	10.1	13	16.3	12	14.0	14	15.9
LA GOMERA	25	3.4	4	7.7	2	2.5	0	0.0	0	0.0	2	3.2	5	5.6	2	2.5	4	5.0	2	2.3	4	4.5
EL HIERRO	21	2.9	2	3.8	5	6.3	0	0.0	4	5.3	0	0.0	4	4.5	2	2.5	2	2.5	0	0.0	2	2.3
No. samples	733		52		79		43		75		62		89		79		80		86		88	

**Table 2 foods-12-00437-t002:** Distribution of sampling by month. Period 2008–2017 (mean values for the whole period are shown).

Month	2008–2017		2008		2009		2010		2011		2012		2013		2014		2015		2016		2017	
	No.	%	No.	%	No.	%	No.	%	No.	%	No.	%	No.	%	No.	%	No.	%	No.	%	No.	%
JANUARY	20	3	0	0	0	0	4	9	3	4	6	10	0	0	1	1	0	0	4	5	2	2
FEBRUARY	60	8	5	10	5	6	8	19	6	8	16	26	6	7	4	5	10	13	0	0	0	0
MARCH	99	14	7	13	7	9	9	21	20	27	14	23	4	4	12	15	10	13	6	7	10	11
APRIL	97	13	8	15	11	14	13	30	8	11	2	3	10	11	12	15	12	15	9	10	12	14
MAY	70	10	5	10	9	11	7	16	14	19	0	0	8	9	2	3	8	10	15	17	2	2
JUNE	43	6	3	6	3	4	2	5	4	5	6	10	7	8	2	3	5	6	5	6	6	7
JULY	53	7	6	12	10	13	0	0	4	5	0	0	4	4	8	10	7	9	10	12	4	5
AUGUST	56	8	2	4	2	3	0	0	2	3	6	10	4	4	6	8	2	3	16	19	16	18
SEPTEMBER	70	10	4	8	8	10	0	0	8	11	11	18	14	16	6	8	11	14	6	7	2	2
OCTOBER	58	8	7	13	14	18	0	0	0	0	1	2	10	11	10	13	6	8	0	0	10	11
NOVEMBER	72	10	3	6	10	13	0	0	0	0	0	0	14	16	9	11	5	6	13	15	18	20
DECEMBER	35	5	2	4	0	0	0	0	6	8	0	0	8	9	7	9	4	5	2	2	6	7
Total	733	100	52	100	79	100	43	100	75	100	62	100	89	100	79	100	80	100	86	100	88	100

**Table 3 foods-12-00437-t003:** Samples by category.

CATEGORY	2008–2017		2008		2009		2010		2011		2012		2013		2014		2015		2016		2017	
	No.	%	No.	%	No.	%	No.	%	No.	%	No.	%	No.	%	No.	%	No.	%	No.	%	No.	%
EXTRA	272	37	29	56	48	61	25	58	27	36	11	18	11	12	21	27	33	41	29	34	38	43
FIRST	290	40	16	31	26	33	16	37	15	20	16	26	29	33	36	46	39	49	50	58	47	53
SECOND	40	5	4	8	5	6	2	5	2	3	4	6	8	9	3	4	2	3	7	8	3	3
UNKNOWN	131	18	3	6	0	0	0	0	31	41	31	50	41	46	19	24	6	8	0	0	0	0
TOTAL	733	100	52	100	79	100	43	100	75	100	62	100	89	100	79	100	80	100	86	100	88	100

**Table 4 foods-12-00437-t004:** Number of active substances analysed by year (2008–2017) considering the detection methods.

	2008	2009	2010	2011	2012	2013	2014	2015	2016	2017
No. active substances	126	125	124	117	138	140	146	158	157	162
LC-MS/MS	101	100	100	92	88	88	88	100	99	103
GC-MS/MS	24	24	23	24	49	51	57	57	57	58

**Table 5 foods-12-00437-t005:** Summary of pesticide residue occurrence in 2008–2017.

Year	2008	2009	2010	2011	2012	2013	2014	2015	2016	2017	2008–2017
N	52	79	43	75	62	89	79	80	86	88	733
R+	49	79	41	70	58	77	72	78	85	86	695
R+ (%)	94.2	100	95.3	93.3	93.5	86.5	91.1	97.5	98.8	97.7	94.8
>MRL	7	6	0	1	1	0	0	0	0	0	15
>MRL (%)	13.5%	7.6%	0%	1.3%	1.6%	0%	0%	0%	0%	0%	2.05%

**Table 6 foods-12-00437-t006:** No. of residues per sample 2008–2017; (N: no. of samples; <MRL: no. of residues < MRL; >MRL: no. of residues > MRL; <MRL/N: ratio of residues < MRL per sample; >MRL/N: ratio of residues > MRL per sample).

	2008	2009	2010	2011	2012	2013	2014	2015	2016	2017	TOTAL
N	52	79	43	75	62	89	79	80	86	88	733
<MRL	127	191	87	133	126	178	162	186	198	217	1.606
<MRL/N	2.4	2.4	2.0	1.8	2.0	2.0	2.1	2.3	2.3	2.5	2.2
>MRL	7	6	0	1	1	0	0	0	0	0	15
>MRL/N	0.13	0.07	0.00	0.01	0.02	0.00	0.00	0.00	0.00	0.00	0.02

**Table 7 foods-12-00437-t007:** Active substances of which residues were detected. 2008–2017; (N: no. of samples; <MRL: no. of residues > LDA < MRL; >MRL: no. of residues > MRL).

Active Substances	N	<MRL	<MRL + (%)	>MRL	>MRL (%)
Bifenthrin	733	22	3.0	0	0.0
Buprofecine	733	82	11.2	0	0.0
Cyfluthrin	733	1	0.1	0	0.0
Cypermethrin	733	13	1.8	7	1.0
Chlorpyrifos	733	522	71.2	0	0.0
Dicofol	262	1	0.4	1	0.4
Dimethoate	733	3	0.4	2	0.3
Spiromesifen	471	1	0.2	0	0.0
Spirodiclofen	333	16	4.8	0	0.0
Etofenprox	733	1	0.1	0	0.0
Ethoxazole	254	7	2.8	0	0.0
Fenamiphos	733	1	0.1	0	0.1
Fenazaquin	733	28	3.8	0	0.0
Fenitrothion	733	4	0.5	5	0.7
Fosthiazate	166	2	1.2	0	0.0
Indoxacarb	690	93	13.48	0	0.0
Iprodione	733	1	0.1	0	0.0
Lambda-cyhalothrin	733	22	3.0	0	0.0
Malathion	733	5	0.7	0	0.0
Tetraconazole	733	1	0.1	0	0.0
Tetradifon	733	1	0.1	0	0.0
Triazophos	733	1	0.1	0	0.0
Acetamiprid	733	85	11.6	0	0.0
Carbendazim	733	1	0.1	0	0.0
Carbofuran	733	2	0.3	0	0.0
Clofentezine	559	30	5.4	0	0.0
Hexythiazox	733	82	11.2	0	0.0
Imazalil	733	502	68.5	0	0.0
Imidacloprid	733	26	3.5	0	0.0
Indoxocarb	174	1	0.6	0	0.0
Iprovalicarb	484	3	0.6	0	0.0
Phenbutaestand Oxide	559	9	1.6	0	0.0
Pencycuron	484	1	0.2	0	0.0
Spinosad	733	8	1.1	0	0.0
Thiabendazole	733	9	1.2	0	0.0
Thiacloprid	733	8	1.1	0	0.0
Trifloxystrobin	484	2	0.4	0	0.0
TOTAL		1595		15	

**Table 8 foods-12-00437-t008:** List of the most frequent active ingredients.

Substance Active	No. of Determinations	No. of Determinations with Residues > LOD	No. of Determinations with Residues > LOD (%)
Chlorpyrifos	733	522	71.2
Imazalil	733	502	68.5
Indoxacarb	690	93	13.5
Acetamiprid	733	85	11.6
Buprofecin	733	82	11.2
Hexythiazox	733	82	11.2
Clofentezine	559	30	5.4
Fenazaquin	733	28	3.8
Imidacloprid	733	26	3.5
Bifenthrin	733	22	3.0
Lambda-cyhalothrin	733	22	3.0
Spirodiclofen	333	16	4.8
TOTAL	8.179	1.510	18.5

## Data Availability

Data is contained within the article or Appendix A.

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
