# Peer review of "Pesticide Residues in Bananas from the Canary Islands"

_foods, 2023, doi:10.3390/foods12030437_

Round 1
Reviewer 1 Report
This study investigated pesticide residues in bananas in Canary Islands from 2008- 2017, which will contribute to assess the use and the residue levels of pesticides. Samples were representative and the number of samples was enough to analyze, so this is a comprehensive and hard work. Introduction and Materials and Methods described in too much detail. Some sentences are too long to be difficult to understand. Some issues should be considered and addressed. The details are as follows:
1. The introduction described in detail but in such an extended way that the introduction is too long. Some sentences (such as in lines 33-37 and 38-43) are too long and some descriptions can be removed. The introduction should be rewritten in a more condensed way to improve its legibility.
2. Line60-69. The mechanism of the relationship between exposure to various pesticides and the carcinogenic effects is far less relevant to this study, so this paragraph can be removed or described in a concise way.
3. Line 74-80. Please described the results of literature in a concise way.
4. Line 81. The semicolon should be replaced with comma. Please check the whole manuscript. Comma should be used between words.
5. Line 102-105. This sentence needs to be redefined. There is no causation relation between residues of active substances not authorized and the pesticides prohibited for banana cultivation in Brazil. The phenomenon should be explained by pesticide abuse or overuse.
6. Line 112-118. Please described these sentences in a concise way.
7. Line 120-121. This paragraph can be removed.
8. Line 134-148. There's no need to describe monitoring programs carried out in the USA, because all results in this study are based on the EU Regulations.
9. Line 153-156. This paragraph can be removed.
10. The Materials and Methods is too long.
11. Line 251-267. This part can be used as supplementary material.
12. Line 310-313. This paragraph can be removed.
13. Line 334. Authors should provide the information of 191 active substances.
14. Table 7. Why dithiocarbamates is listed separately? LC-MS/MS and GC-MS/MS are detection methods, so the heading of column should be detection method.
15. Line 342. Table 8 is missing.
16. Line 363-364. Abbreviation should be used when the word occurs firstly. GC-MS/MS and LC-MS/MS occurred in line 345.
17. Determination of pesticide residues by GC-MS/MS. Authors should provide more information about the sample preparation, such as the amount of MgSO4, NaCl, anhydrous sodium acetate for QuEChERS and the speed of centrifugation. Besides, some conceptual descriptions about the GC-MS/MS procedure in lines 379-391 should be removed.
18. Determination of pesticide residues by LC-MS/MS. The determination of pesticide residues by GC-MS/MS and LC-MS/MS can be combined one part. Some information are repeated such as sample extraction (line 392-396 and 437-441), identification and confirmation criteria.
19. Please perform your discussion in line with your important findings.
20. Figures need to be improved.
21. Line 469-472. Although the value of samples with Residues >MRL decreased radically, the no. of samples with Residues>ADL<MRL did not decreased. Thus, this result cannot come to the conclusion of a better rationalisation of pesticide use by operators.
22. Line 511. …….shows a decreasing variation in the period studied (2006 - 2015): 1.4% in 2006; 0.4% in 2009; 0.7% in 2012 and 0.3% in 2005. Please rewrite this sentence in chronological order. The residues > MRL in 2009 is 0.4% from Coordinated Community Programme, but in this study is 7.6% in 2008. Why results show such a big difference? I don’t think the trend of both are similar.
Author Response
The responses to the reviewer can all be found in the attached file.

Reviewer 2 Report
This paper presented an investigation of pesticide residues in banana samples collected from different regions of Canary Islands. The purpose was to provide some information for the establishment of several MRL values of certain pesticides in bananas. A lot of data were obtained and analyzed. The content of this research could be useful for the safety of banana consumption in this city. However, several revisions should be conducted. The “Introduction” is too tedious which should be concise. For example, the importance of this study should be emphasized and some general introduction of pesticide and crop (not banana) should be deleted. The authors should use more figures to present the results instead of tables, which could show the content more clearly. The existed figures should be replaced by the clearer ones. The format of the tables should be unified. To make the presentation of this study more understandable, the conclusion should be reorganized.
Author Response

(The authors gave the same response as above.)

Reviewer 3 Report
L. 82. buprofezin?
L. 115 Please refer to and evaluate the results of the last years. The EFSA report is available up to 2020.
L.166 Please use LOQ consistently in the whole manuscript
L. 167 specify the meaning of in a „significant % of the samples” What is the basis for the reference?
Table 2: revise the title
L180-183 Please correct the sentence
Table 3 L226-232 The principle of the sampling plan is good. However, the number of samples taken from the different islands is far lower than would be required to draw such conclusions as postulated. ..samples, as far as possible, is distributed uniformly throughout the year, so that in each month there is a significant number of samples to establish a reliable correlation with the pesticides detected in each month.
When 6-16 samples are taken annually such a statement cannot be justified. (evan 20-35 samples are not sufficient)
Please revise the sentence and set realistic objectives.
Taking into account the 4 quality categories the number of samples seems to be insufficient for such differentiation. Pls. reconsider your statements.
L. 236-237 please specify what is the basis for claiming a significant relationship based on 20 samples
Tables 4 and 5. Please consider the above points
L. 297-298 Random sampling has its own strict criteria. Can you be more specific on how these were considered and met?
L.315 banana fingers?
Table 6 is not necessary as the last line of table 5 contains the same information.
L. 356: multi-residue analytical offer: use method or analyses
L.437: consider describing the extraction, cleanup, and identification procedures once
L.462-465 Pls. clarify the sentence
Table 11. Pls clarify the abbreviations used in the table and bring them in line with those used in the text
LMR?
482-486: What was the commodity in which the quoted high level of violation was observed especially within the EU?
L. 489 different validated methods of analyses should provide comparable results. Your statement implies that either your or the US data are not reliable due to different methods. Revised the sentence
492-495: This stamen is rather questionable since EU laboratories are not using the same standardized methods; reporting the uncertainty should not affect the numerical value of the residues measured
L.504-511 The community monitoring program comprised different crops and consequently different residues. This is not the appropriate basis for comparison.
All your arguments on the comparability of results (L487.541) are practically outside the scope of this article and can be omitted without losing the value of the work.
What is m.a. for in Figure 2?
Figure 3 Pls prepare the legend in English or provide a translation
Regarding the conclusions
Table 11. Define ADL
L608-609. Clarify the cases above the MRL

Author Response

(The authors gave the same response as above.)

Round 2
Reviewer 2 Report
This paper can be accepted.